# Determination of Lactoferrin in Camel Milk by Ultrahigh-Performance Liquid Chromatography-Tandem Mass Spectrometry Using an Isotope-Labeled Winged Peptide as Internal Standard

**DOI:** 10.3390/molecules24224199

**Published:** 2019-11-19

**Authors:** Xia Li, Zengmei Li, Enmin Xu, Ling Chen, Hua Feng, Lu Chen, Ligang Deng, Dongliang Guo

**Affiliations:** 1Institute of Agricultural Quality Standards and Testing Technology Research, Shandong Academy of Agricultural Sciences, 202 Gongyebeilu Road, Jinan 250100, Shandong Province, China; Lisa-fsd@163.com (X.L.); lizengmei78@163.com (Z.L.); wangjialinmm@163.com (H.F.); chenludemail@163.com (L.C.); 2Shandong Veterinary Drug Quality Inspection Institute, 68 Huaicun Street, Jinan, Shandong Province, China; xuenm@126.com (E.X.); 13708935340@163.com (L.C.)

**Keywords:** lactoferrin, camel milk, ultrahigh-performance liquid chromatography-tandem mass spectrometry, heparin affinity chromatography

## Abstract

An ultrahigh-performance liquid chromatography-tandem mass spectrometry method was developed and validated for the determination of lactoferrin in camel milk based on the signature peptide. The camel lactoferrin was purified by heparin affinity chromatography and then used to screen tryptic signature peptides. The signature peptide was selected on the basis of sequence database search and identified from the tryptic hydrolysates of purified camel lactoferrin by ultrahigh-performance liquid chromatography and quadrupole time-of-flight tandem mass spectrometry. The pretreatment procedures included the addition of isotope-labeled winged peptide and the disposal of lipids and caseins followed by an enzymatic digestion with trypsin. Analytes were separated on an Acquity UPLC BEH 300 C18 column and then detected on a triple-quadrupole mass spectrometer in 7 min. The limits of detection and quantification were 3.8 mg kg^−1^ and 11 mg kg^−1^, respectively. The recoveries ranged from 74.5% to 103.6%, with relative standard deviations below 7.7%. The validated method was applied to determine the lactoferrin in ten samples collected from Xinjiang Province.

## 1. Introduction

Lactoferrin is an iron-binding glycoprotein that was first identified in 1939 in bovine milk [1], and in 1960 it was isolated from human milk by Johannson [2]. It belongs to the family of transferrins, together with serum transferrin, ovotransferrin, and melanotransferrin [3]. The molecular weight of lactoferrin is about 80 kDa, and it consists of a single polypeptide chain folded into two globular lobes with the ability to bind two iron atoms [4,5]. Lactoferrin plays an important role in innate defense and exhibits multifunctional activities, including maintaining iron levels, as well as antimicrobial activities, antioxidant activities, immunomodulation, and suppression of tumor growth and metastasis [6,7,8,9]. In addition, lactoferrin has potential use in the treatment of osteoarthritis and other inflammatory diseases [10,11]. These factors make it a supplementary functional food and stimulate increasing interest in exploiting the therapeutic value of lactoferrin [8,12].

Lactoferrin is widely distributed in mucosal secretions of mammalian species, such as tears, saliva, and bile, and is in highest abundance in milk and colostrum [6]. The lactoferrin found in camel milk and colostrum is at concentrations of 0.18–2.48 mg mL^−1^ and 0.59–5.10 mg mL^−1^ [13,14], respectively, higher than in cows, goats, and buffaloes [13,15]. Camel milk was frequently reported to have higher antimicrobial activity and biological value than those of bovine milk [15,16]. It is significantly more heat resistant than cow and buffalo milk lactoferrins [15]. The annual output of camel milk is about 2.9 million tons all over the world [17], and camel milk is one of the main components of the human diet in many parts of the world. However, the concentration of camel lactoferrin is period-dependent and is significantly dependent on season [13,14]. The concentration changes of lactoferrin may affect the nutritional value and biological functions of camel milk. For nutritional assessment and quality control, it is necessary to establish reliable analytical methods for the determination of lactoferrin in camel milk.

Some analytical techniques such as spectrophotometry [3], high-performance liquid chromatography (HPLC) [13], and immunochemical assay [13,15,18] have been used for the determination of camel lactoferrin. Spectrophotometry and HPLC methods have inherently poor precision and insufficient resolution, because the different forms of lactoferrin (native, apo-, and holo-lactoferrin) may affect their physicochemical properties such as color, particle charge, and rheological behavior [19]. The immunodiffusion technique has the advantages of easy operation and simple instrumentation, but this method has poor precision. Furthermore, the preparation of antigen and antibody is complicated and expensive, and it is not suitable for the testing of large quantities of samples. Enzyme-linked immunosorbent assay (ELISA) is more selective and sensitive; the ELISA kit has been used for bovine lactoferrin determination [20,21,22]. The commercially available ELISA kit is not suitable for camel lactoferrin determination, however, because the target protein cannot bind the antigen and antibody in the kit. To our knowledge, there is no simple, robust, and accurate detection method that can be used for the rapid quantification of camel lactoferrin.

Liquid chromatography-tandem mass spectrometry (LC-MS/MS), a popular detection method that is emerging because of its high sensitivity, specificity, and reliability, has increasingly contributed to expanding protein analysis based on entire proteins or tryptic peptides. Some major proteins in bovine milk such as casein, beta-lactoglobulin, and alpha-lactalbumin have been investigated by LC-MS/MS method [23,24,25]. A similar analytical method was also developed for the determination of bovine lactoferrin in dairy products [12]. However, it is not suitable for camel lactoferrin determination because the signature peptide used for MS/MS analysis represents bovine lactoferrin, and the amino acid sequence in camel lactoferrin is not the same [3].

The aim of this study was to develop and validate a simple, robust, and accurate method for the rapid quantification of camel lactoferrin by LC-MS/MS. The camel lactoferrin was purified by heparin affinity chromatography and then digested with trypsin. A signature peptide having higher response and ionization efficiency is selected as the representative of camel lactoferrin protein. The actual isotopically labeled internal standard of the lactoferrin was synthesized according to the tryptic winged peptide. Subsequent analysis was performed by LC-MS/MS in the multiple reaction monitoring (MRM) mode under positive ionization mode. The contents of camel lactoferrin were calculated on basis of the equimolar relationship between lactoferrin protein and lactoferrin signature peptide. Finally, the validated method was applied to determine the camel lactoferrin contents of ten camel milk samples.

## 2. Results and Discussion

### 2.1. Purification of Camel Lactoferrin

Since lactoferrin can bind anionic compounds such as heparin and DNA, these materials have been used to purify lactoferrin [7]. The use of heparin-Sepharose affinity chromatography for the purification of lactoferrin from human whey was first described by Bläckberg and Hernell [26], and this method was used for the purification of lactoferrin from bovine milk [27]. On the basis of the elution profile of human or bovine lactoferrin eluted from a heparin-Sepharose column [26,27], the elution gradient of 1.0 M sodium chloride in 10 mM sodium phosphate buffer (pH 7.0) were used to remove impurities and to collect camel lactoferrin. The purified camel lactoferrin was analyzed by SDS-PAGE. The single protein band with an estimated molecular mass of 75 kDa was apparent, indicating that the purification of camel lactoferrin was accomplished (Figure 1); the purified lactoferrin was used for subsequent research.

### 2.2. Selection and Synthesis of Signature Peptide Standard for Camel Lactoferrin

For the development of an MRM-based quantitative method for the determination of camel lactoferrin, it is crucial to choose a suitable signature peptide representing camel lactoferrin. Factors such as specificity of amino acid sequences, high efficiency of ionization, strong intensity of MS signal, and prevention of contamination with susceptible amino acid such as cysteine and methionine are essential requirements in the selection of signature peptides [12,25]. Trypsin, which is the commonly used protease for protein digestion and tryptic peptides, tends to be doubly charged. The specific peptides for camel lactoferrin were chosen by comparing the theoretical and endogenous peptides from tryptic camel lactoferrin.

The complete amino acid sequence of camel lactoferrin has been studied and is available in the publicly accessible protein database [3]. The theoretical tryptic peptide of camel lactoferrin was obtained by computational prediction using Waters Biolynx software. The endogenous tryptic peptides were acquired by analysis of the tryptic digests of purified camel lactoferrin and camel milk. By UHPLC-Q-TOF analysis and sequence database search, the peptide DVTVLDNTDGK corresponding to residues 545–555 of camel lactoferrin was selected and synthesized as specific biomarkers of camel lactoferrin because of its specificity, high signal intensity, and sensitivity. The candidate peptide was doubly charged with *m*/*z* of 588.7, which is in good agreement with the theoretical values. The proposed glycosylation sites in camel lactoferrin were Asn^233^, Asn ^366^, Asn^518^, and Asn^575^ [3], and selection of peptide avoided the above four sites. Mass transitions were selected as *m*/*z* 588.7 → 649.2 and *m*/*z* 588.7 → 762.5 from the production ion mass spectra of the synthetic peptide DVTVLDNTDGK, which correspond to b4 and b5 fragment ions, respectively (Figure 2). The specificity and selectivity of the synthesized peptide DVTVLDNTDGK was confirmed by analyzing the camel milk after trypsin cleavage.

### 2.3. Optimization and Synthesis of Isotopically Labeled Signature Peptide and Internal Standard

The method for the quantitation of camel lactoferrin consisted of a sample preparation procedure to remove lipids and caseins, which was followed by a UHPLC-MS/MS analysis of the whey protein isolate. Since casein is the major protein in camel milk, comprising about 52–87% of the total proteins [28], removal of casein can prevent its interference in the process of tryptic digestion and then reduce the usage of trypsin and improve the efficiency of enzyme digestion. The recovery of the isolation procedure and proteolytic rate were variable between samples and experiments. Furthermore, the ionization efficiency and the presence of other peptides and matrix components tend to affect the accuracy of this method. In order to minimize the isolation recovery, ionization efficiency, and digestion variability, a winged peptide was used as internal standard. The sequence of the winged peptide is DVAFVKDVTVL*DNTDGKNTEQWAK. It is composed of a stable isotope-labeled signature peptide and six or seven amino acid residues along with the sequence of camel lactoferrin at each end. The sequence of the stable isotope-labeled signature peptide is DVTVL*DNTDGK. There is only one isotope-labeled amino acid in the signature peptide and internal peptide, but it is sufficient to distinguish the isotope-labeled one from its native counterpart by MS via a 7 Da mass shift. In addition, it can lower the cost of synthesis of isotopically labeled internal standard and signature peptide. A similar approach with one isotopically labeled amino acid in signature peptide and internal peptide has been applied to measure phosphoproteins from cell lysates and thyroglobulin in serum and plasma [29,30]. The stable isotope-labeled peptide is chemically identical to its native counterpart formed by proteolysis, and its mass transitions were optimized as *m*/*z* 592.4 → 649.1 and *m*/*z* 592.4 → 769.4 from the product ion mass spectra, which correspond to b4 and b5 fragment ions, respectively (Figure 2). The signature peptide, stable isotope-labeled signature peptide of camel lactoferrin, showed similar chromatographic performance and good linear response during the UHPLC-MS/MS analysis (Figure 3). The tryptic digestion efficiency of camel lactoferrin and the internal standard were evaluated using the corresponding tryptic amount and compared with the known amount of camel lactoferrin or the internal standard. The digestion efficiency was more than 94.3% and 93.8% for camel lactoferrin and its synthetic internal standard, respectively, when they were spiked into the mobile phase. The consistency of digestion efficiency indicated that the synthetic internal standard could mimic the analytical behavior of intact camel lactoferrin.

### 2.4. Method Validation

#### 2.4.1. Specificity

The specificity of the signature peptide was assessed by online BLAST search in UniProt (www.uniprot.org) and NCBI (www.ncbi.nlm.nih.gov). The results of BLAST search show that the amino sequence of signature peptide only exists in lactoferrin of camel milk. When human milk, bovine milk, buffalo milk, and goat milk were analyzed by UHPLC-MS/MS, ion pairs of 588.7 → 649.2 and m/z 588.7 → 762.5 (*m*/*z*) were not found in these samples. The analytical results further confirm the specificity of the signature peptide and the established method. Comparison of the retention time of the synthetic peptide standards and the selected signature peptide from tryptic samples showed that both of them had a sharp and symmetric peak at 4.09 ± 0.02 min. All of these results indicated the specificity of the signature peptide for camel lactoferrin. 

#### 2.4.2. Linearity, Sensitivity, and Repeatability

The method exhibited good linearity between the area ratio *y* (analyte to internal standard) versus the concentration ratio *x* (analyte to internal standard) in the range of 10–500 nM. The typical linear regression equation was *y* = 0.1176*x* − 0.00288. The correlation coefficient (*r*) of the standard curve was greater than 0.999. The LOD and LOQ were 3.8 mg kg^−1^ and 11.0 mg kg^−1^, respectively, which were estimated to be the lowest concentration corresponding to 3 and 10 times the S/N ratio, respectively. This result suggested that the proposed method was sensitive enough for detecting camel lactoferrin in camel milk. The repeatability expressed as the RSD was obtained from the results from multiple measurements (*n* = 6) of each sample. The RSD was 3.2–7.8%, which demonstrated that the developed UHPLC-MS/MS method was reproducible.

#### 2.4.3. Recovery and Precision

A recovery experiment was performed to evaluate the accuracy of the method. Intraday and interday variations, as well as RSDs of the peak areas, were calculated to express the precisions. Table 1 provides mean recoveries and RSDs of the analytes. The spiking recovery rates were 74.5–103.6% with the RSD of 6.4–7.7%. The RSDs of intra- and interday precision were determined as 5.5–8.9% and 3.3–10.1%, respectively. All of the results demonstrated that the current method had good recovery and precision, and that it was able to satisfy the requirements for the quantification of camel lactoferrin in camel milk.

### 2.5. Method Application

The validated method was applied to determine the lactoferrin in 10 samples collected from Xinjiang Province. All samples were digested with trypsin and subjected to UHPLC-MS/MS analysis according to the aforementioned procedures. The selected signature peptide from camel lactoferrin and its isotope-labeled analog from spiked internal standard were successfully detected in the tryptic cleavage products of all samples (a typical chromatogram is shown in Figure 4). The amounts of camel lactoferrin in liquid milk were 62–651 mg kg^−1^, consistent with literature reports [13,14]. The lactoferrin is generally considered as particularly rich in colostrum or milk from camel. An extremely high lactoferrin content of 5.10 mg mL^−1^ was reported in one camel colostrum sampled 2 d after parturition, as compared with about 0.50 mg mL^−1^ in bovine colostral milk [13]. The concentration of lactoferrin significantly decreased after 8 days of milking, whereas the values were still higher than those in bovine milk [18]. The variability of lactoferrin in camel milk was determined by the lactation stage, species of camel, as well as seasonal and geographic conditions.

## 3. Materials and Methods

### 3.1. Materials

Ammonium bicarbonate (NH_4_HCO_3_), calcium chloride (CaCl_2_), sodium chloride (NaCl), sodium hydroxide (NaOH), disodium hydrogen phosphate (Na_2_HPO_4_), and acetic acid, which were of analytical grade, were purchased from Sinopharm Chemical Reagent Co., Ltd. (Shanghai, China). Lactoferrin from bovine milk (>85%), Tris (2-carboxyethyl) phosphine hydrochloride (TCEP), and iodoacetamide (IAA) were obtained from Sigma–Aldrich (St. Louis, MO, USA). Sequencing-grade modified trypsin was from Promega Corporation (Madison, WI, USA). Methanol, acetonitrile, and formic acid of HPLC grade were purchased from Fisher Chemicals (Fair Lawn, NJ, USA). Ultrapure water was obtained from a Milli-Q gradient water system (Millipore, Bedford, MA, USA). HiTrap^TM^ Heparin HP was obtained from GE Healthcare Bio-Science AB (Uppsala, Sweden). Ultracel^®^-50K centrifugal filters were obtained from Merck Millipore Ltd. (Tullagreen, Carrigtwohill, Ireland).

### 3.2. Synthetic Peptide Standards

The signature peptide DVTVLDNTDGK (corresponding to amino acid residues 545–555 of camel lactoferrin), stable isotope-labeled signature peptide DVTVL*DNTDGK, and internal standard DVAFVKDVTVL*DNTDGKNTEQWAK were synthesized by Sangon Biotech (Shanghai) Co., Ltd. (Shanghai, China). The stable isotope-labeled ^13^C, ^15^N-Fmoc-D-leucine gave a total molecular mass shift of 7 Da from the nonlabelled peptide. All the peptide standards were synthesized with purity of more than 95%.

### 3.3. Purification of Camel Lactoferrin

For the purification of camel lactoferrin, the camel milk was defatted by centrifugation at 4000× *g* for 20 min at 4 °C. The skim milk was adjusted to pH 4.6 with acetic acid at 20 °C and kept at 4 °C for 30 min, and then it was centrifuged at 8000× *g* for 30 min to remove casein. The suspension was transferred to a centrifugal filter (Ultracel^®^-50K) and centrifuged at 3000× *g* for 60 min at 4 °C to remove low molecular weight proteins (less than 50 kDa). The residual solution was adjusted to pH 7.0 and then applied to a heparin affinity column (HiTrap^TM^ Heparin HP) equilibrated with 10 mM sodium phosphate solution (pH 7.0). The elution gradient was 0.5 M sodium chloride in 10 mM sodium phosphate buffer (pH 7.0) to remove impurities, and 2.0 M sodium chloride in 10 mM sodium phosphate buffer (pH 7.0) for camel lactoferrin. All buffers were prepared with deionized water.

### 3.4. Electrophoresis

Sodium dodecyl sulfate-polyacrylamide gel electrophoresis (SDS - PAGE) was performed with a 5% stacking gel and a 12% separating gel containing 0.1% SDS, followed by running in 0.125 M Tris-HCl at pH 6.8 and 0.38 M Tris-HCl buffers at pH 8.8, respectively. The proteins were stained with Coomassie Brilliant Blue 250 and destained with a methanol (30%) and acetic acid (7%) solution. The absorbency measured at 280 nm and an extinction coefficient of 84540 M^−1^ cm^−1^ were used to calculate the concentration of camel lactoferrin [3].

### 3.5. Preparation of Tryptic Hydrolysates

Prior to tryptic hydrolysis, the camel milk was adjusted to pH 4.6 with acetic acid, and then it was centrifuged at 8000× *g* for 30 min to remove casein and fat. Aliquots of 100 μL of whey were spiked with 100 μL of 1 μM stable isotope-labeled internal standard DVAFVKDVTVL*DNTDGKNTEQWAK and then mixed with 180 μL of 50 mM NH_4_CO_3_. The mixtures were reduced by adding 15 μL of 500 Mm TCEP solution at 50 °C for 30 min. An alkylation was performed in the presence of 45 μL of 500 mM IAA solution at room temperature for 30 min in the dark. Subsequently, 10 μL of 100 mM CaCl_2_ solution and enough trypsin was added, and then it was incubated at 37 °C overnight. The reaction was terminated by the addition of 10 μL of formic acid, and then the mixture was diluted to 1 mL and with ultrapure water. After the filtrate was passed through a 0.22 mm nylon filter, it was analyzed by LC-MS/MS.

### 3.6. Liquid Chromatography

Chromatographic separation was performed on an ACQUITY Ultra Performance Liquid Chromatography System (Waters, Milford, MA, USA) consisting of a cooled autosampler and a column oven. The system was equipped with a Waters Acquity UPLC BEH C18 column (100 mm × 2.1 mm I.D., 1.7 μm, 300 Å). Mobile phase A was Milli-Q water containing 0.1% ammonia, and mobile phase B was acetonitrile. A binary gradient system with a flow rate of 0.3 mL min^−1^ was established. The gradient steps were as follows: 10% B for 0–1 min, linear increase from 10% to 90% B for 1–4 min, and holding for 6.5 min; and 10% B for 6.5–7 min followed by re-equilibration of the column for 3 min. The injection volume was 5 μL, and the total run time for each injection was 7 min. The effluent from the UHPLC system was directed into the electrospray ion (ESI) source of the mass spectrometer (MS).

### 3.7. Mass Spectrometry

Search and identification of signature peptide for camel lactoferrin was performed on a Xevo G2-S QTof mass spectrometer equipped with ESI source (Waters, Milford, MA, USA). The instrument was operated in the electrospray positive ion (ESI^+^) mode. The conditions were as follows: capillary voltage of 3.0 kV, sampling cone voltage of 30 V, extraction cone voltage of 4.0 V, source temperature of 100 °C, desolvation temperature of 450 °C, cone gas of 50 L h^−1^ undefined nitrogen; and desolvation gas of 800 L h^–1^ nitrogen. The data acquired in MSE continuum mode were processed using MassLynx 4.1 (Waters) and analyzed using ProteinLynx Global Server version 2.5 software (PLGS 2.5, Waters). The search parameters were trypsin enzyme, fixed modification site of carboxymethyl, and a maximum of one missed cleavage. The specificity of the signature peptide selected for camel lactoferrin was confirmed by BLAST program in the National Center for Biotechnology Information (NCBI) database.

A Xevo TQ-S triple quadrupole mass spectrometer (Waters, Milford, MA, USA) equipped with ESI source was used for all quantitative data. Sample introduction and ionization was done in the positive mode between 200 and 1200 *m*/*z*. The capillary voltage was 3.50 kV and the cone voltage was 15 V. Nitrogen was used as desolvation gas at a flow rate of 850 L h^−1^. The desolvation gas temperature was 500 °C, and the source temperature was 130 °C. Argon was used as collision gas at the pressure of 3 × 10^3^ mbar. Mass transitions monitored in the method were *m*/*z* 588.7 → 762.5, 588.7 → 649.2 for the peptide DVTVLDNTDGK, and 592.3 → 769.4, 592.3 → 649.2 for the peptide DVTVL*DNTDGK. Data were acquired in MRM mode, and the scheduled MRM function was used as the acquisition method to ensure enough acquisition points (at least 12 points for each peak).

### 3.8. Method Validation

The following parameters were studied for method validation: Specificity, detection (LOD), quantification (LOQ) limits, linearity, recovery, precision, and accuracy. The specificity was demonstrated by comparing the retention time of the synthetic signature peptide standard and natural peptide from tryptic samples. The method LOD and LOQ at which the signal-to-noise ratio had to be at least 3 for the LOD and 10 for the LOQ were determined. For the determination of range and linearity of the method, six signature peptide standards containing a fixed concentration of stable isotope-labeled signature peptide in the range of 10–500 nM were prepared and analyzed by the internal standard method. The recovery of the present method was evaluated by employing the standard addition method. To evaluate the recoveries of the preparation method, purified camel lactoferrin was spiked into fresh camel milk at low, medium, and high concentrations. The recovery was calculated according to the following equation: Recovery (%) = (measured concentration—the original level)/spiked concentration × 100%. Intraday and interday precision and accuracy of the method were studied. Intraday precision was determined by analyzing one sample six times in succession on the same day. For interday precision, the same samples were prepared and determined on five different days. Precision was calculated as RSD in percentage.

## 4. Conclusions

In this study, we developed and validated a UHPLC-MS/MS method for the quantitative determination of camel lactoferrin on the basis of the signature peptide derived from the tryptic hydrolysates of camel lactoferrin. The method has two steps: The sample preparation procedures, which include the addition of an isotopically labeled peptide as internal standard followed an enzymatic digestion with trypsin, and the determination of camel lactoferrin by UHPLC-MS/MS analysis. The signature peptide was selected by comparing the endogenous and theoretical peptides from tryptic camel lactoferrin. The specificity, sensitivity, repeatability, and precision of this method demonstrated that the current method was able to satisfy the requirements for the quantification of camel lactoferrin in camel milk. The present method was successfully applied to determine lactoferrin in camel milk and whole milk powder. To the best of our knowledge, this is the first time that camel lactoferrin was quantitatively analyzed through a UHPLC-MS/MS method.

## Figures and Tables

**Figure 1 molecules-24-04199-f001:**
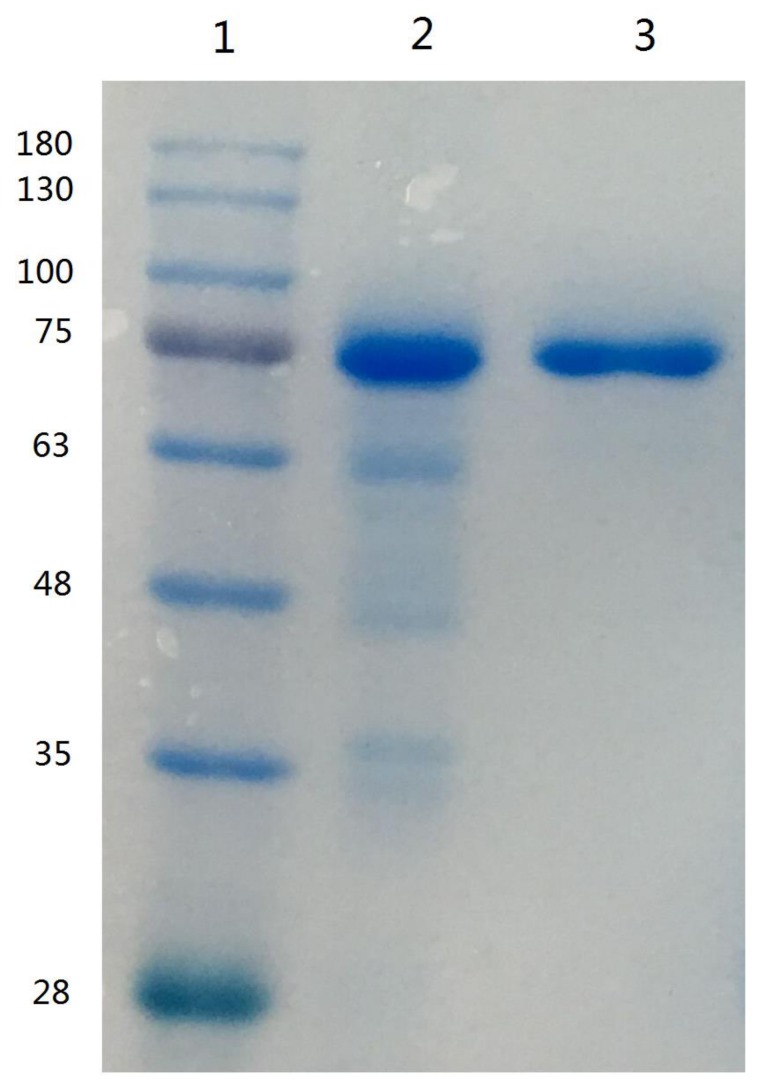
SDS-PAGE of camel lactoferrin obtained from camel milk. Lane 1, molecular weight standards; lane 2, lactoferrin from bovine milk; lane 3, purified camel lactoferrin.

**Figure 2 molecules-24-04199-f002:**
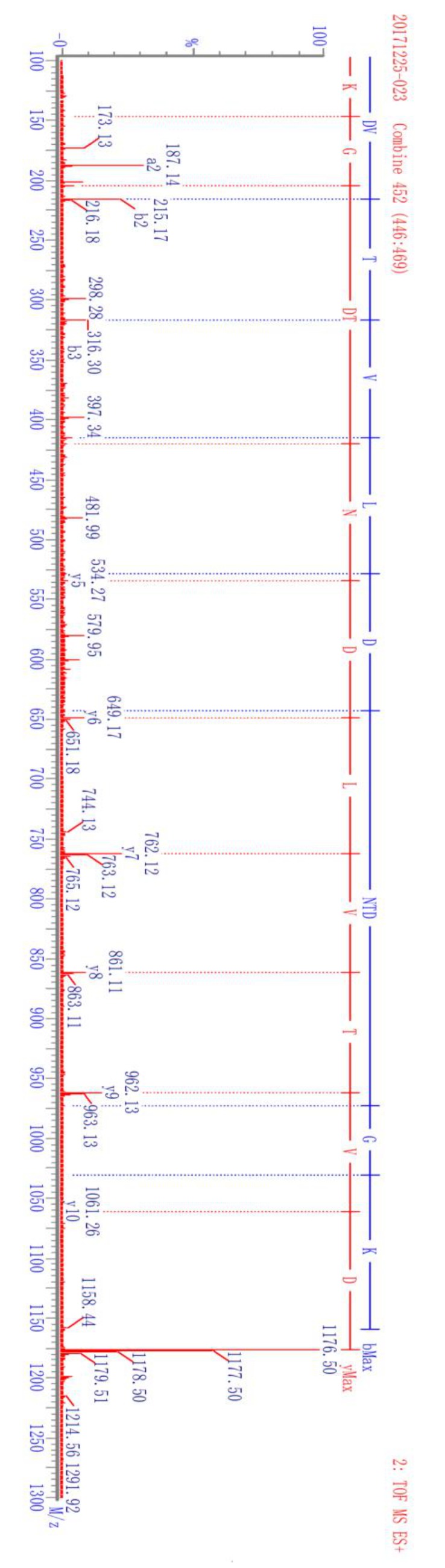
Fragment ions camel lactoferrin signature peptide DVTVLDNTDGK and its corresponding isotope-labeled analog DVTVL*DNTDGK.

**Figure 3 molecules-24-04199-f003:**
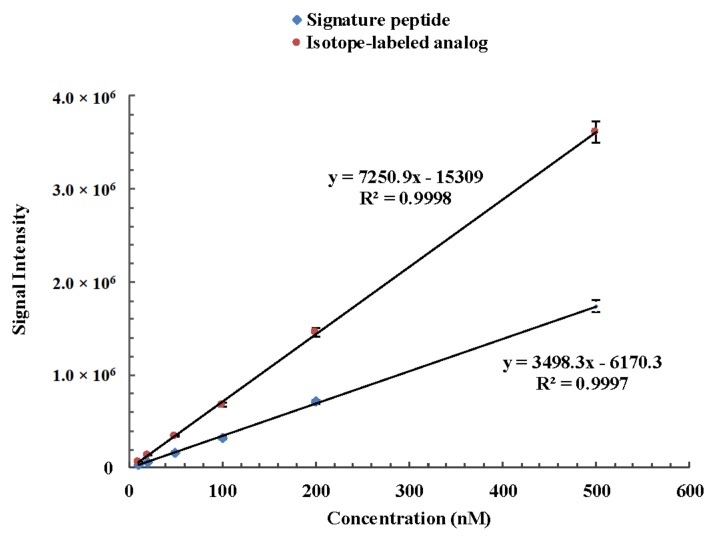
Linear response of camel lactoferrin signature peptide DVTVLDNTDGK and its corresponding isotope-labeled analog DVTVL*DNTDGK during the UHPLC-MS/MS analysis.

**Figure 4 molecules-24-04199-f004:**
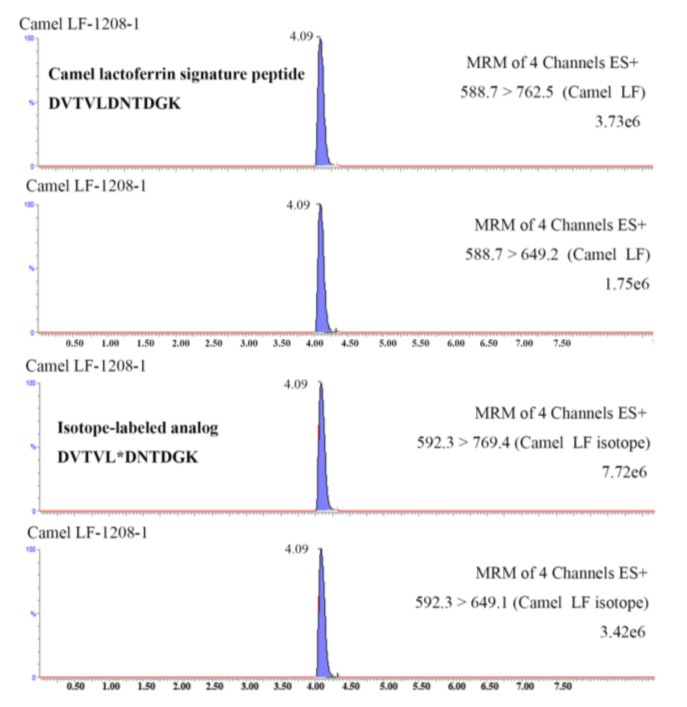
UHPLC-MS/MS chromatograms of camel lactoferrin signature peptide DVTVLDNTDGK (B) and its corresponding isotope-labeled analog DVTVL*DNTDGK in a tryptic dairy sample.

**Table 1 molecules-24-04199-t001:** Spiked recovery of the UHPLC-MS/MS method for determination of camel lactoferrin (n = 6).

Original Level (mg/100 g)	Spiked Level (mg/100 g)	Determined Level (mg/100 g)	Recovery Rate (%)	RSD%
6.2 ± 0.5	6.0	10.6 ± 0.3	74.5 ± 5.7	7.7
29.3 ± 1.0	30.0	60.4 ± 2.3	103.6 ± 7.6	7.3
65.1 ± 4.5	60.0	123.4 ± 3.6	97.1 ± 6.0	6.4

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
