# Peer review of "Determination of Lactoferrin in Camel Milk by Ultrahigh-Performance Liquid Chromatography-Tandem Mass Spectrometry Using an Isotope-Labeled Winged Peptide as Internal Standard"

_molecules, 2019, doi:10.3390/molecules24224199_

Round 1
Reviewer 1 Report
The authors proposed a method for the Determination of Lactoferrin in Camel Milk by Ultrahigh-Performance Liquid Chromatography-Tandem Mass Spectrometry Using an Isotope-Labeled Winged Peptide as Internal Standard. The manuscript is well present and the content is interesting, therefore It is recommended for publication after some minor correction.
81 format problem
The authors should provide all standard purity.
Figure 2 and 3 have very low resolution.
It’s not clear if the UPLC method optimized or if the authors just adapted already exist methods.
Reviewer 2 Report
Please make Figure 2 more visible. Increase the size, to compromise the space make Figure 1 smaller.
Line 118: For mass transition notation, please use an arrow
Line 176: Please add error bars for Figure 3
Line 190: Table 1. Units should be “mg/ 100 g” not “Mg/100 g”
Author Response

This manuscript is a resubmission of an earlier submission. The following is a list of the peer review reports and author responses from that submission.